# Automatic Active Lesion Tracking in Multiple Sclerosis Using Unsupervised Machine Learning

**DOI:** 10.3390/diagnostics14060632

**Published:** 2024-03-16

**Authors:** Jason Uwaeze, Ponnada A. Narayana, Arash Kamali, Vladimir Braverman, Michael A. Jacobs, Alireza Akhbardeh

**Affiliations:** 1Department of Computer Science, Rice University, Houston, TX 77005, USA; 2Department of Diagnostic and Interventional Imaging, The University of Texas Health Science Center at Houston, Houston, TX 77030, USA; 3Department of Radiology and Radiological Science and Sidney Kimmel Comprehensive Cancer, Johns Hopkins University School of Medicine, Baltimore, MD 21287, USA; 4The University of Texas MD Anderson Cancer Center UT Health Houston Graduate School of Biomedical Sciences, Houston, TX 77030, USA

**Keywords:** multiple sclerosis, dimensionality reduction, multiparametric MRI, lesion segmentation

## Abstract

Background: Identifying active lesions in magnetic resonance imaging (MRI) is crucial for the diagnosis and treatment planning of multiple sclerosis (MS). Active lesions on MRI are identified following the administration of Gadolinium-based contrast agents (GBCAs). However, recent studies have reported that repeated administration of GBCA results in the accumulation of Gd in tissues. In addition, GBCA administration increases health care costs. Thus, reducing or eliminating GBCA administration for active lesion detection is important for improved patient safety and reduced healthcare costs. Current state-of-the-art methods for identifying active lesions in brain MRI without GBCA administration utilize data-intensive deep learning methods. Objective: To implement nonlinear dimensionality reduction (NLDR) methods, locally linear embedding (LLE) and isometric feature mapping (Isomap), which are less data-intensive, for automatically identifying active lesions on brain MRI in MS patients, without the administration of contrast agents. Materials and Methods: Fluid-attenuated inversion recovery (FLAIR), T2-weighted, proton density-weighted, and pre- and post-contrast T1-weighted images were included in the multiparametric MRI dataset used in this study. Subtracted pre- and post-contrast T1-weighted images were labeled by experts as active lesions (ground truth). Unsupervised methods, LLE and Isomap, were used to reconstruct multiparametric brain MR images into a single embedded image. Active lesions were identified on the embedded images and compared with ground truth lesions. The performance of NLDR methods was evaluated by calculating the Dice similarity (DS) index between the observed and identified active lesions in embedded images. Results: LLE and Isomap, were applied to 40 MS patients, achieving median DS scores of 0.74 ± 0.1 and 0.78 ± 0.09, respectively, outperforming current state-of-the-art methods. Conclusions: NLDR methods, Isomap and LLE, are viable options for the identification of active MS lesions on non-contrast images, and potentially could be used as a clinical decision tool.

## 1. Introduction

Multiple sclerosis (MS) is the most common demyelinating disease in humans and it affects both white matter and gray matter of the central nervous system (CNS; brain and spinal cord). The etiology of MS is not completely understood [1,2]. Magnetic resonance imaging (MRI) is the most common radiologic modality for noninvasive visualization of MS pathology in the CNS. MRI plays a crucial role in both diagnosis and patient management. A hallmark of MS is the presence of hyperintense lesions on T2-weighted (T2W), including T2-weighted Fluid Attenuation by Inversion Recovery (FLAIR), and proton-density-weighted (PDW) MRI. However, not all hyperintense lesions seen on T2W MRI are active. Identification of active lesions is essential for patient management [3,4,5,6,7]. Regionally compromised blood–brain barrier (BBB) is believed to be a pathologic feature of active lesions. The compromised BBB is the result of repeated venular inflammation [8]. The compromised BBB allows the leakage of gadolinium (Gd)-based contrast agents (GBCAs) from the vasculature into the brain parenchyma, which results in reduced T1 relaxation time because of the paramagnetic nature of GBCA. Consequently, on MRI, active lesions appear hyperintense on post-contrast T1-weighted (T1W) images (contrast enhancement). An association between Gd enhancement and clinical activity in MS patients has been reported [8,9]. Moreover, the number of volumes of enhancing lesions may be important in the evaluation of treatment efficacy [9,10].

Recent studies have reported that GBCA administration results in long-term accumulation of gadolinium (Gd) in various tissues [11]. While there is no definite evidence demonstrating any physiologic effect of accumulated Gd, the Food and Drug Administration issued a cautionary note that “clinicians should limit GBCA use to circumstances in which additional information provided by the contrast agent is necessary, and assess the necessity of repetitive MRIs with GBCAs” [12]. In response to these concerns, a number of techniques for identifying active lesions on unenhanced scans have been proposed [13]. Perhaps the most widely accepted method for detecting active lesions is through subtraction MRI. These methods are mainly based on the subtraction of MR images acquired at two different time points to detect expanding lesions that are considered to represent lesion activity [14]. However, all these techniques have been introduced to identify active lesions, but not quantitatively compare with the ground truth, post-contrast T1-weighted images. More recently deep learning (DL) has started making inroads in automatically identifying active lesions on unenhanced scans [15].

A number of automatic and semi-automatic techniques for identifying and segmenting enhancing lesions in MS have been reported [5,10,16,17,18]. As described by Coronado et al. [17], these methods have limitations that include the need for specialized MRI pulse sequence, extensive pre-processing, minimizing false lesion classification etc. Therefore, more recently, deep learning was used to delineate enhancing lesions in MS [16,17,19]. These DL methods are mainly based on convolutional neural networks (CNNs).

The majority of MS segmentation methods, including DL, are based on multiparametric MRI. While multiparametric-based segmentation is shown to provide excellent results, it also suffers from a few weaknesses. These include the following: (1) high dimensional feature space that increases the computational complexity, (2) ineffectiveness in explicitly capturing the spatial relationship between voxels, (3) inability to effectively capture nonlinear relationships between different voxels, since it linearly combines voxel intensities of images with different contrasts, (4) noise within MRI can compromise segmentation quality, and (5) requiring registration of multiparametric images. Many of these problems can be potentially alleviated using nonlinear dimensionality reduction (NLDR) techniques.

Machine learning is broadly divided into supervised and unsupervised learning methods [20]. Supervised learning requires labeled data created by experts to establish the ground truth. The labeling by experts could be expensive in time and cost. While CNNs are excellent at extracting global features and detecting large lesions, they may have limitations in extracting multi-scale local features, thus making it hard to detect active lesions, which may vary in size. Data-intensive tasks, like segmentation, often suffer from the curse of dimensionality, which suggests that the required sparsity and sample size needed to achieve optimal results increase with the dimensionality of the data [21]. To overcome these limitations, in this preliminary study, we have applied an unsupervised NLDR method for segmenting active lesions in MS.

Non-linear dimensionality reduction is an unsupervised machine learning technique [22,23,24]. Unlike linear dimensionality reduction (LDR), NLDR captures complex nonlinear data structures that are prominent in brain MR images. In a high-dimensional feature space, NLDR uses local features from nearest-neighbor graphs to create a lower-dimensional subspace. NLDR requires far fewer images to train than DL-based methods. This work investigates how NLDR techniques can be used to identify active MS lesions.

The objective of this study was to demonstrate that unsupervised NLDR methods outperform supervised machine learning methods in identifying active lesions. To accomplish this, we introduce key differences between LDR and NLDR. Subsequently, we establish the motivation behind applying NLDR to brain MRI data. Then, we provide details for the MR imaging dataset and performance evaluation metrics used in this work. Furthermore, we compare NLDR methods to current state-of-the-art methods for active MS lesion identification. Lastly, we discuss the potential limitations of this work.

## 2. Materials and Methods

### 2.1. Dimensionality Reduction

For combining multiparametric MRI data while maintaining inherent data structure, also called the manifold, we used dimensionality reduction (DR). A manifold is a subspace that allows the visualization of curves and surfaces through multiple coordinate systems, or charts [23]. Intuitively, a manifold is any object that can be “charted”. DR is achieved by either keeping a small subset of the most relevant information and features or by finding a smaller subset of data points [24]. DR maps high dimensional dataset *X*, with *D* dimensions, to a lower dimensional dataset *Y*, with *d* dimensions, such that *d* < *D* and *X* can be represented by *d* points.

Dimensionality reduction can be divided into two categories, LDR and NLDR, which make assumptions about *X* lying in a linear and nonlinear subspace, respectively, [25,26]. While both methods can find the lowest number of data points that are able to represent the structure of *X*, there are key differences between LDR and NLDR that are briefly described below.

### 2.2. Linear Dimensionality Reduction

LDR might reduce the complexity of high-dimensional data through a linear mapping of all data points. With LDR, we assume that the subspace that best fits *X* is linear. LDR has been used in several healthcare applications because it is highly adaptable and effective in extracting features from large high dimensional datasets [27,28,29]. An example of LDR is multidimensional scaling (MDS).

#### Multidimensional Scaling

MDS finds a linear subspace, Y, that best represents the high dimensional dataset, X, by minimizing the cost function below
(1)$$\sum_{ij}\left(|\right|x_{i}-x_{ij}\left|\right|-\left|\right|y_{i}-y_{ij}{\left||\right)}^{2}$$
such that, given the *i*, *j*-th components of *X* and *Y*, $\left|\right|x_{i}-x_{ij}\left|\right|$ and $\left|\right|y_{i}-y_{ij}\left|\right|$ are the respective Euclidean distances. To minimize this cost function, MDS uses eigenvalue decomposition defined as
(2)$$\begin{array}{c} B=X^{T}*X, \end{array}$$
(3)$$\begin{array}{c} M=[V_{1},V_{2},\ldots Vd], \end{array}$$
(4)$$\begin{array}{c} Y=M*X, \end{array}$$
where *V* are the eigenvectors, *M* is a feature matrix containing the first d largest eigenvalues, and *B* is the pairwise distances matrix of *X*.

Recently, Knezek et al. [27] used DR to assess various learning approaches to improve motivation amongst young learners. These authors applied MDS to add meaning and understanding to educational game play, and its effect on children’s learning performance. This work goes beyond DR for conducting statistical analysis. Malone et al. [28] proposed a Clinical Sustainability Assessment Tool (CSAT) for grouping health concepts, collected from 64 healthcare and research professionals, into several domains (e.g., engaged staff, planning, and monitoring) to create a meaningful tool for improving sustainable healthcare service delivery. Both studies used MDS to remove redundant information from high-dimensional data and create helpful insights. Although MDS effectively reduces high dimensional data, it struggles to uncover the true manifold of non-linear data, thus motivating the use of NLDR.

### 2.3. Non-Linear Dimensionality Reduction

We used NLDR, to uncover the true manifold of multiparametric brain MR images by reconstructing a single embedded image. While LDR assumes that the data lie in a linear subspace, NLDR can perform more intricate embeddings of nonlinear data [22]. Brain MRI is one of the primary methods for diagnosing MS, often requiring multiple image sequences to identify and characterize pathology. Brain MRI sequences, such as T2-weighted (T2W) and T1W images, provide different contrast information that aids patient management in MS. Since brain MR images share a nonlinear subspace, we employed NLDR for segmentation of active lesions. This is accomplished by reducing the dimensionality of MRI sequences into lower dimensions while preserving contrast information. Specifically, we used two NLDR algorithms, isometric feature mapping (Isomap) [22,30] and locally linear embedding (LLE) [31] to segment MS lesions using multiparametric MRI. The following sections will briefly describe how Isomap and LLE create a low-dimensional representation of high-dimensional data.

#### 2.3.1. Isometric Feature Mapping

Isomap maps a high-dimensional dataset, *X*, to a lower-dimension dataset, *Y*, while retaining maximum geometric information. To reconstruct *Y*, Isomap first uses a K nearest neighbor graph *G* [32] to calculate geodesic distances (GDs) [33,34] of all data points of *X*. This creates a distance matrix, such that every data point, $x_{i}$, is connected to its K nearest neighbors, $x_{ij}$. Isomap then utilizes MDS to map all data points in *X* to a lower dimension [22,35]. To preserve the distance of *X*, MDS employs eigenvalue decomposition of the pairwise distance matrix. Specifically, we used the K neighbors classifier for the implementation of *G* and MDS. We used both *G* and MDS to reconstruct multiparametric brain MRI into a single embedded image. To maintain the manifold of *X*, MDS minimizes the error between pair-wise distances defined below
(5)$$\sum \left(|\right|x_{i}-x_{ij}\left|\right|-\left|\right|y_{i}-y_{ij}{\left||\right)}^{2}$$
where $\left|\right|x_{i}-x_{ij}\left|\right|$ and $\left|\right|y_{i}-y_{ij}\left|\right|$ are the geodesic distances between data points in higher and lower dimensions, respectively. Isomap finally uses MDS mappings to create *Y*, a single image embedding *X*. The steps to create Isomap embeddings are shown in Figure 1.

We used two shortest path algorithms to construct pairwise distance matrices, crucial for Isomap geodesic distance calculations. In Tenenbaum et al. [22], Isomap was introduced using shortest path algorithm, a recursive method which attempts to create M while holding the following recurrence relation [36,37], to create a geodesic distance matrix. Floyd Warshall is a recursive method which attempts to create M while holding the following recurrence relation
(6)$$d_{ij}^{k}=\left\{\begin{array}{cc} w_{ij} & \;\mathrm{if},\;k=0\\ \min\left(d_{ij}^{k-1},d_{ik}^{k-1},d_{kj}^{k-1}\right), & \;\mathrm{if},\;k>0 \end{array}\right.$$
where $w_{ij}$ and $d_{ij}^{k}$ represent the initialized and shortest distance lengths between points *i* and *j*, respectively, for all of data points *X*. Dijkstra’s is another popular shortest distance algorithm [37], which first selects a starting data point, labeled as the source, and finds the shortest path from the source to all other data points of *X*. According to Wang [37], Dijkstra is defined by the following relation
(7)$$D_{ij}=\min(D_{ij},D_{ij}+dist\left(x_{j},x_{k}\right)$$
where $D_{ij}$ is the current distance from data point *i* to *j*, $dist(x_{j},x_{k})$ is the distance between two abutting data points, and matrix M is created by recursively finding the pairwise distances of all data points of *X*. Due to the simplicity of the Floyd Warshall algorithm, it is often preferred over Dijkstra’s algorithm. Set to default in the scikit learning library, the best algorithm was automatically selected at run time, to assume optimal performance.

#### 2.3.2. Locally Linear Embedding (LLE)

Similar to Isomap, LLE creates a mapping from *X* to a lower dimension while preserving its local properties. LLE assumes that embeddings can be created through small regions of *X*, such that any data points, $x_{i}$, and its neighbors, $x_{ij}$, lie on or near a linear patch of *X*. By assuming that the manifold is approximately linear, LLE reconstructs the properties of the data by calculating the sum of K nearest neighbors for each data point. We used *G* in order to calculate the Euclidean distances for all data points in *X*. LLE identifies coordinates of *Y* or $y_{i}$ that best align with the embedding of *X* [31].

To create an embedding of *X*, LLE characterizes the local geometry of all patches of *X*, as linear coefficients, and reconstructs each data point from its neighbors. The reconstruction error is defined as
(8)$$\epsilon \left(W\right)=\sum_{n=1}^{n}\left|\right|x_{i}-\sum_{j=1}^{k}w_{ij}x_{ij}{\left|\right|}^{2},$$
which is constrained by $w_{ij}=0$ when $x_{i}$ has no neighbors (i.e., $x_{ij}=0$) and $\sum_{j=1}^{k}w_{ij}=1$. To perform embedding of *X* in *Y*, LLE chooses the best coordinates of *Y* to minimize the embedding error defined as
(9)$$\phi \left(Y\right)=\sum_{n=1}^{n}\left|\right|y_{i}-\sum_{j=1}^{k}w_{ij}y_{j}{\left|\right|}^{2}.$$The mapping from *X* to *Y* in LLE is carried out by minimizing the reconstruction and embedding error in Equations (Equation 2) and (Equation 5).

#### 2.3.3. Applications of Non-Linear Dimensionality Reduction

Recently in healthcare, NLDR techniques have been incorporated to extract important features from medical data. Sharma et al. [38] introduced a framework for thyroid abnormality detection, applying various dimensionality reduction techniques to ultrasound and histopathological datasets. Results based on NLDR methods outperformed current state-of-the-art computer-assisted diagnostic systems. In this work, LLE and Isomap exceeded other DR techniques, such as principal component analysis and singular value decomposition, in extracting features from ultrasound data, demonstrating that NLDR is able to reconstruct nonlinear ultrasound imaging data. We chose to use NLDR methods, LLE and Isomap, to segment active MS lesions. NLDR has been shown to outperform LDR methods in extracting features from imaging datasets [21,25,38].

In Figure 2, we show how NLDR compares to the LDR method, MDS. Dimensions of the Swiss roll were successfully reduced to a lower dimension by using LLE and Isomap, meaning both pairwise distances of all data points and the geometric shape of the Swiss roll were preserved in the embedded images. While successfully creating a lower dimensional representation, MDS failed to unfold the manifold while maintaining its underlying structure. This suggests that LLE and Isomap effectively understand complex manifolds and can be applied to real-world data. We further validated this hypothesis on multiparametric brain MRI data and showed how LLE and Isomap are able to segment active MS lesions.

### 2.4. Patients

The MRI data were acquired as a part of CombiRx, a multi-center, phase 3, double-blinded, randomized clinical trial (Clinical Trial Identifier: NCT00211887) [39] that was supported by the National Institute of Health (NIH). Table 1 summarizes the demographics and clinical data of the clinical trial cohort at baseline. The main objective of the CombiRx clinical trial is to evaluate if treating MS patients with a combination interferon-β1a (IFNB) and glatiramer acetate (GA) results in a better patient outcome compared to treating them with these two drugs separately. Only patients with the relapsing–remitting (RRMS) phenotype are included in this trial. Our dataset included MRIs from 46 randomly selected patients. All patients provided signed written consent and this study was approved by the local IRB. All patients had MRI-detectable MS brain lesions and were administered GBCA during the MRI scan. In total, 6 subjects were excluded due to image misregistration issues, bringing the total number of subjects to 40.

### 2.5. Image Dataset

Anonymized brain MR images from 46 subjects with relapsing–remitting MS were used in this work. MRIs were acquired on Philips (Best, The Netherlands) or General Electric (Milwaukee, USA), or Siemens (Erlangen, Germany) scanners operating at either 1.5T or 3T field strengths. The images included proton density (PDW), T2W, 2D FLAIR, and pre- and post-contrast T1W with voxel dimensions of (0.94 mm × 0.94 mm × 1.5 mm). All images were reviewed by two neuro MRI experts (more than 30+ years of experience) [40,41]. The MRI protocol and the acquisition details were provided elsewhere. Image preprocessing methods including co-registration, bias field correction and intensity normalization were applied [39].

The target lesions of this study are active white matter lesions. Grey matter lesions are not included because they rarely show enhancement. Also, grey matter lesion detection requires specialized MRI sequences such as double inversion recovery (DIR) sequence [42] and the CombiRx MRI did not include DIR imaging.

### 2.6. Multiparametric Brain MRI

We used LLE and Isomap to differentiate different tissue types and identify active lesions from multiparametric brain MRI. Specifically, we used PDW, FLAIR, T2W, and pre-contrast T1W as our high-dimensional dataset, and applied NLDR methods in identifying active MS lesions in brain MRI. Post-contrast T1W images were excluded from the input but were used for validation of the embedded image-based active lesion identification. The ground truth used was subtracted pre- and post-contrast T1-weighted image, referred to as the subimage. Lesions identified in subimages are labeled as active. To obtain lesion regions, we performed binary thresholding on all the ground truth and embedded images. Lower dimensional images, created by LLE and Isomap embeddings, were verified to include MS lesions by one of the authors (P.A.N).

As summarized in Figure 3, active lesion segmentation was accomplished by (1) embedding of brain MRI images through LLE and Isomap applications, (2) calculating of the subimage, and (3) generating binary images followed by thresholding to isolate lesions. The performance of LLE and Isomap was evaluated using thresholded binary images. To quantitatively evaluate the performance of LLE and Isomap, we used Dice similarity to measure the overlap between lesion boundaries in embedded and ground truth lesions. By converting LLE and Isomap embedding to binary images, we assigned “1” to pixels related to active MS lesions, and a zero to all other pixels.

### 2.7. Performance Evaluation

To describe the overlap between the embedded and ground truth binary images, we use DS
(10)$$DS=2\frac{A\cap B}{A+B}$$
where *A* is the ground truth and B represents lesions identified on the binary thresholded images.
(11)$$DS=\frac{2TP}{2TP+FN+FP}$$

Equation (Equation 11), as shown in Figure 4, TP, FP, FN, represent the true positive, false positive, and false negative sets of pixels [43], based on the overlap between A and B. Thus, if no overlap is found, DS = 0, or 1 if A and B perfectly overlap. Brain MR image resolution and the nearest neighbor value, K, were the input parameters to the model. We empirically evaluated the performance of LLE and Isomap for optimizing different input parameters.

## 3. Results

In order to choose the optimal input parameters for Isomap and LLE, we evaluated their sensitivity to various input resolutions and nearest neighbor values, K.

Due to high computational times in testing Isomap and LLE on original brain MR images, we initially reduced the input resolutions. However, Figure 5 shows that reducing the input resolution of LLE and Isomap degrades both methods’ ability to provide accurate DS scores due to the blurring of active lesion boundaries. Our results suggest that Isomap and LLE are able to segment active lesions given 128^2^ input images but achieve optimal performance on the original 256^2^ input resolution. As a result, we chose 256^2^ as the input resolution for LLE and Isomap to obtain accurate DS measurements.

As shown in Figure 6, both Isomap and LLE are insensitive to K ≥ 100. However, LLE appears to have a minor fluctuation of dice scores across k-values, and thus is more sensitive to k-values compared to Isomap, which is more stable.

To evaluate Isomap and LLE on all MS patients (n = 40), we chose K to be 100. With the chosen input resolution and K value, Isomap and LLE successfully segmented active lesions from brain MRI. Figure 7 shows single embedded images reconstructed by Isomap and LLE, while Figure 8 shows the segmentation performance of Isomap and LLE on all subjects. Isomap achieved a median DS of 0.78 ± 0.09, compared to LLE with a median DS of 0.74 ± 0.1. These results need to be further validated on a larger sample size.

## 4. Discussion

To the best of our knowledge, this is one of the first implementations of Isomap and LLE for gadolinium (Gd) enhancing (active) lesion segmentation. We utilized NLDR techniques to combine brain MR images into an embedded image and identify lesions. Our results show, based on DS, that both Isomap and LLE were excellent at identifying active lesions, with Isomap providing slightly better results relative to LLE. Based on our results, LLE and Isomap could potentially reduce the need for the administration of contrast agents and outperform previous state-of-the-art active lesion segmentation DL methods.

Administration of GBCA [7,44,45] is still used for identifying active lesions in MS. With recent observations about GBCA retention in various tissues, different methods have been proposed to identify Gd-enhancing lesions, either by reducing the amount of GBCA administered or eliminating it. However, the majority of published methods, particularly those based on DL, utilized multiparametric MRI and supervised learning. These methods have their own disadvantages. In this preliminary study, we attempted to overcome these disadvantages by using unsupervised techniques, Isomap, and LLE. Our results suggest that NLDR methods if confirmed on a larger sample size, can be deployed in a clinical setting.

Multiple MRI studies are reported on the detection of active lesions on unenhanced scans. The results of subtraction images are compared with post-T1W images to assess the accuracy. However, they did not perform lesion segmentation [7,14,46,47,48]. Thus, it is very difficult to compare our results with these published ones. On the other hand, there are existing AI-based clinical applications for active lesion identification. In [7], Rudie et al. identified active MS lesions on non-contrast images using a computer-assisted-detection (CAD)-based system. This work demonstrated that it is possible to perform real-time assessments of non-contrast images to determine the necessity of contrast administration. However, this work requires manual annotations of MS lesions to achieve optimal performance. LLE and Isomap achieved excellent performance in identifying active MS lesions without need for manual annotated trained images as input. Identifying active lesions without contrast agents and less supervision could potentially improve patient safety and reduce the cost of clinical care.

Based on Coronado et al. [17], our work appears to outperform current state-of-the-art (SOTA) DL methods in segmenting active lesions in brain MRI. However, this work included GBCA imaging while we excluded post-contrast T1W images from testing. Current methods for active MS lesion segmentation use deep convolutional neural networks (CNNs) [17,19,40,41] to segment active lesions from brain MRI. Coronado et al. achieved a mean dice similarity score of 0.77 which is considered to be excellent. Our method offered a very comparable score of 0.78. Our method offers a number of advantages over the CNN method, which typically needs a large amount of annotated imaging data by experts, which could be very time-consuming and not feasible for large data. Although this work achieved dice similarity performance as shown in this preliminary study, our method is an unsupervised machine learning technique and does not require using a subset of data for training and the entire dataset could be used as a validation set.

Instead of using manual lesion annotations to guide reconstruction, LLE and Isomap use local image features from K nearest neighbor graphs to reconstruct high dimensional inputs in a lower dimensional space. This suggests NLDR performance is dependent on local topological features [22,32]. Other segmentation studies of breast [25], and ischemic lesions in brain MRI [49], used Isomap and LLE to reconstruct multiparametric MRI data with fewer than 30 subjects.

Akhbardeh et al. [25] used DR to embed multiple breast MR images and segment breast lesions. The performance of linear and nonlinear DR methods was compared to determine the best method for breast tissue segmentation. Both LLE and Isomap outperformed LDR methods, principal component analysis and MDS, and resulted in excellent similarity to post-contrast T1-weighted images that are consistent with our results. In this work, we exploited the robustness of NLDR methods for segmenting smaller embedded MS lesions compared to larger breast lesions.

To segment tissue at risk in stroke patients and quantify the amount of salvageable brain tissue, Parekh et al. [49] integrated multiparametric stroke MRI data using LLE and Isomap, demonstrating high similarity to radiological gold standards. However, this study was limited to animal MRI data, and thus needs to be further validated on human subjects. With the application of NLDR in early cerebral ischemia detection, this work shows NLDR methods as potential solutions for other diseases that require multiparametric MRI data to perform accurate diagnosis. Similarly, with brain MRI data, Park [50] used Isomap to quantify the shape of brain MRI and detection of Alzheimer’s Disease. This work revealed that Isomap can delineate the true shape of brain MR images, distinguish between AD and normal manifolds, and potentially be used for brain MRI time-series analysis.

There are potential limitations in our study. For example, there is increased computational complexity, in both Isomap and LLE, when reconstructing embedded images. This is largely due to shortest path distance and eigenvalue calculations. This limitation could be overcome using a subset, “landmarks”, of high dimensional data points to reduce the computations required for both Isomap and LLE [51,52,53,54].

Another limitation is that our sample size was 40 subjects. Further validation is needed on a larger and independent sample size that is drawn from a different distribution. Lastly, this work does not measure the sensitivity of LLE and Isomap to different lesion sizes. Investigating and addressing active MS lesion identification performance as a function of lesion sizes could further improve the performance of LLE and Isomap.

## 5. Conclusions

In conclusion, we demonstrated unsupervised learning techniques, LLE and Isomap, are able to capture the manifold of multiparametric MRI data and segment active MS lesions. Although both methods exhibited excellent performance, Isomap slightly outperformed LLE and showed more consistent results in the sensitivity analysis of input parameters. LLE and Isomap potentially could assist in the clinical management of MS patients, and reduce the usage of GBCAs for identifying active lesions in brain MRIs.

## 6. Patents

A.A. and M.A.J. have patent, “Multiparametric non-linear dimension reduction methods and systems related thereto”, US Patent 9,256,966.

## Figures and Tables

**Figure 1 diagnostics-14-00632-f001:**
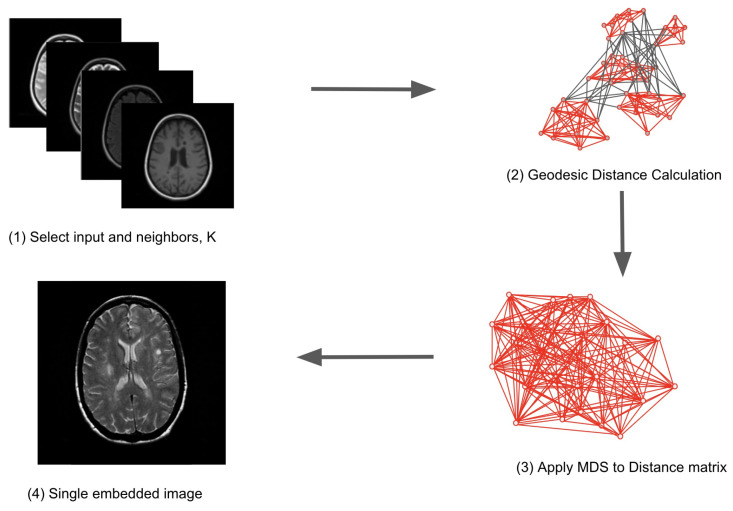
The Isomap algorithm can be summarized in four steps: (1) select a nearest neighbor algorithm, (2) calculate geodesic distances for all data points (3) apply multidimensional scaling to geodesic distance matrix, (4) output single embedded image.

**Figure 2 diagnostics-14-00632-f002:**
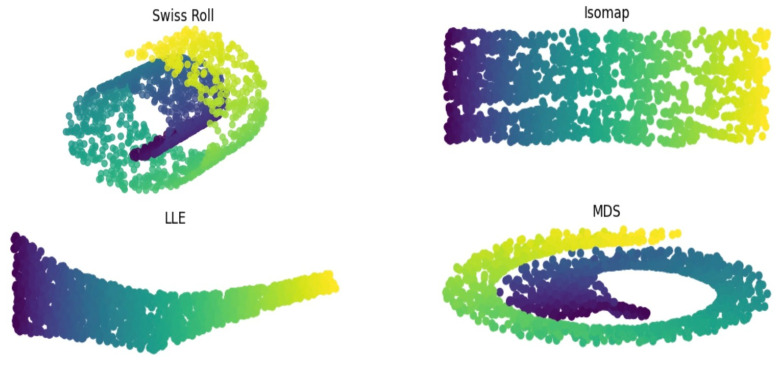
Embedding of the 3-dimensional Swiss roll to 2 dimensions using MDS, LLE, and Isomap. Neighborhood size, k, was set to 12 for all methods. Both NLDR methods, LLE and Isomap, unfolded the Swiss roll in lower dimensions while the linear DR method, MDS, failed to unfold and preserve the manifold. Results that best retained the shape of the Swiss roll were derived by Isomap.

**Figure 3 diagnostics-14-00632-f003:**
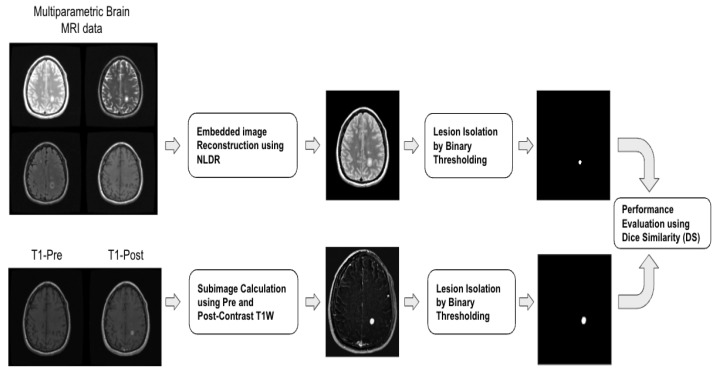
Multiparametric input brain MRI integration framework for embedding reconstruction. (**Top row**) Four brain MRIs are inputted and then integrated into a single embedded image using NLDR methods. The NLDR maps are thresholded for lesions. (**Bottom row**) T1-pre and post are subtracted. Next, active lesions are masked to obtain the ground truths binary masks. At the end, the two binary images are compared using dice similarity (DS) score, for the evaluation of proposed methods.

**Figure 4 diagnostics-14-00632-f004:**
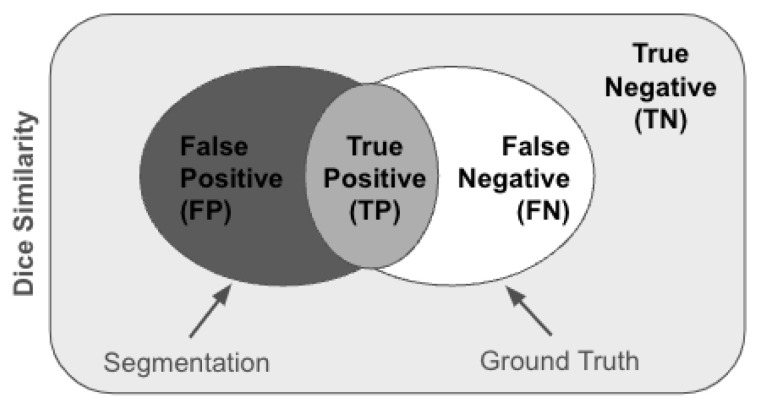
Dice Similarity is used to describe the overlap between embedded and ground truth active lesions.

**Figure 5 diagnostics-14-00632-f005:**
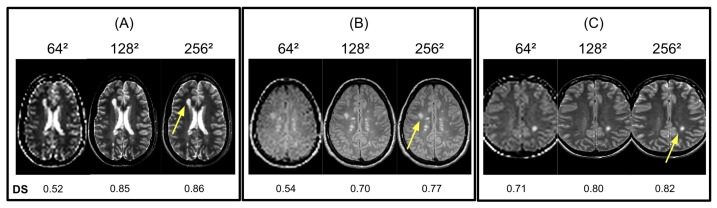
Examples of LLE algorithm sensitivity to different image matrix sizes and resolutions. We compared the active lesion (yellow arrow) in the embedded image to ground truth lesions using DS scores. (**A**–**C**) As shown in 64^2^ images, (bottom row) DS was significantly lower. However, no significant differences were observed between 256^2^ and 128^2^.

**Figure 6 diagnostics-14-00632-f006:**
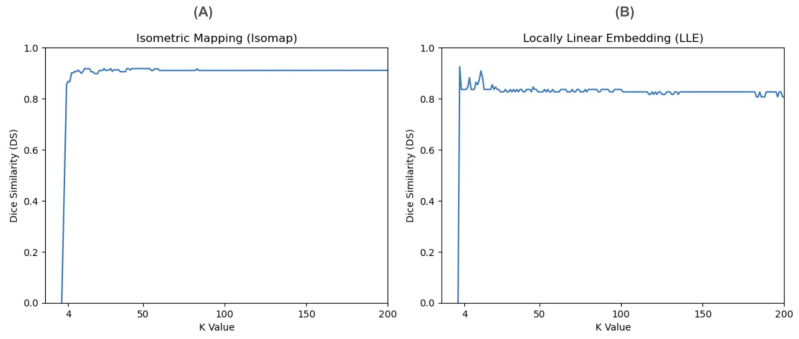
Evaluation of the sensitivity of NLDR methods to control parameters (K; neighborhood size): (**A**) Isomap (**B**) LLE.

**Figure 7 diagnostics-14-00632-f007:**
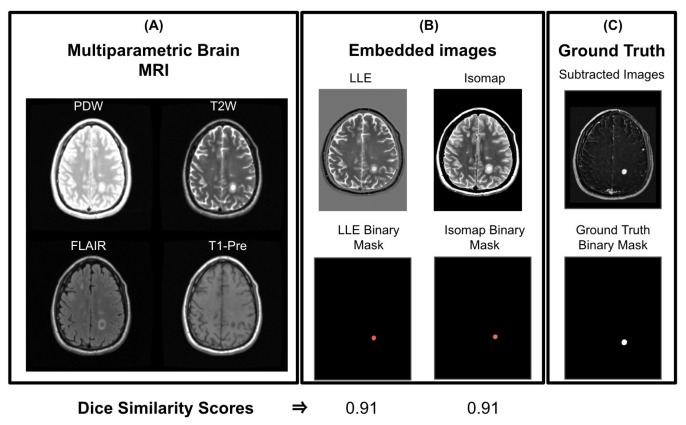
(**A**) Multiparametric brain MRI data as input to nonlinear dimensionality reduction (NLDR) methods. (**B-upper**) Output embedded images from locally linear embedding (LLE) and isometric feature mapping (Isomap). (**B-lower**) The resulting binary masks are shown. (**C-upper**) The ground truth (contrast subtraction image). (**C-lower**) Corresponding binary image for ground truth, used to evaluate methods using dice similarity scores.

**Figure 8 diagnostics-14-00632-f008:**
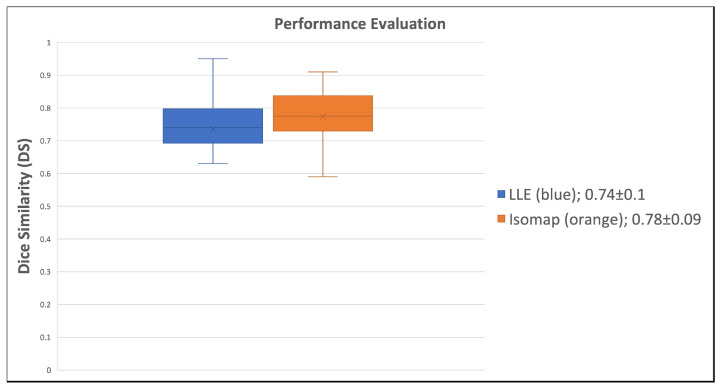
The performance of nonlinear dimensionality reduction methods, locally linear embedding (LLE) and isometric feature mapping (Isomap), are shown in the box plot. Isomap marginally outperformed LLE with a median and standard deviation for dice similarity (DS) score, respectively, as 0.78 ± 0.09 and 0.74 ± 0.09.

**Table 1 diagnostics-14-00632-t001:** Demographic and clinical data acquired as part of CombiRx randomized clinical trial.

	Demographics and Clinical Data on the CombiRx Cohort	
Age (yrs)		37.7 ± 9.7
Female/Male (ratio)		72/28
	Caucasian	87.6
Race (%)	African American	7.2
	Other	5.2
	Hispanic	6.3
Ethnicity (%)	Non-Hispanic	89.5
	Other	4.3
Symptom Duration (yrs)		4.8 ± 5.6

## Data Availability

Data will be made available upon request, and availability is determined by the institutional guidelines.

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
