# Peer review of "Automatic Active Lesion Tracking in Multiple Sclerosis Using Unsupervised Machine Learning"

_diagnostics, 2024, doi:10.3390/diagnostics14060632_

Round 1

Reviewer 1 Report

Comments and Suggestions for Authors

This study focuses on identifying automatically identifying active lesions on brain MRI, without the administration of contrast agent using NLDR, LLE and Isomap methods.

Authors must include research questions taken for this study. There is no existing literatures, there must be a detail on research gaps in existing literature and how the current study overcome the gaps.  

The work clearly addresses the motivation for GBCA-free active lesion detection and outlines the objective and methodology in detail

There is no detailed explanation on data, include the size and source of the patient dataset? Were specific lesion types targeted?

There is no comparison to other methods, address whether the study include comparisons with other non-contrast lesion detection methods, not just deep learning?

Highlight whether there are any limitations to the NLDR approach, such as specific types of lesions it might miss?

Are there are any study envision translating these findings into clinical practice? If so , it can be added to improve the quality of the work

Authors must address, whether the study consider the sensitivity and specificity of the methods for different lesion sizes and locations?

Reports promising results with LLE and Isomap achieving higher Dice similarity scores than current state-of-the-art methods.

The work is well concluded  with the potential clinical utility of these methods as a decision tool.

This is a promising study with the potential to significantly impact the diagnosis and treatment of MS. Addressing the above feedbacks for discussion could further strengthen the article and provide valuable insights for future research.

Comments on the Quality of English Language

Grammer and typos can be thoroughly checked.

Reviewer 2 Report

Comments and Suggestions for Authors

The article entitled “Automatic Active Lesion Tracking in Multiple Sclerosis Using Unsupervised Machine Learning” is well-written and, from my point of view, would be of interest for the readers of Diagnostics. In spite of this and before its publication, I would suggest authors to perform the following changes:

In the introduction a more in-depth explanation of the aims of the research is required. Also, a brief paragraph describing the whole structure of the manuscript would be worthy.

In materials and methods section, lines 85-87 it is said: “Dimensionality reduction can be divided into two categories, Linear DR and nonlinear DR (NLDR), which make assumptions about X lying in a linear and nonlinear subspace, respectively.” Some bibliographical references about such statements would be welcome.

Please introduce any information about Arnoldi decomposition, not only citing a library that implements it.

In line 208 it is said: “MRIs were acquired on Philips or GE or Siemen scanners operating at either 1.5T or 3T” please if posible, introduce scanners models.

Figure 3 is too small. Please, split it in two.

Round 2

Reviewer 1 Report

Comments and Suggestions for Authors

Authors have addressed the issues. Statisfied